# Dynamics of Gut Microbiota and Short-Chain Fatty Acids during a Cycling Grand Tour Are Related to Exercise Performance and Modulated by Dietary Intake

**DOI:** 10.3390/nu16050661

**Published:** 2024-02-27

**Authors:** Manuel Fernandez-Sanjurjo, Javier Fernandez, Pablo Martinez-Camblor, Manuel Rodriguez-Alonso, Raquel Ortolano-Rios, Paola Pinto-Hernandez, Juan Castilla-Silgado, Almudena Coto-Vilcapoma, Lorena Ruiz, Claudio J. Villar, Cristina Tomas-Zapico, Abelardo Margolles, Benjamin Fernandez-Garcia, Eduardo Iglesias-Gutierrez, Felipe Lombó

**Affiliations:** 1Department of Functional Biology (Physiology), University of Oviedo, 33006 Oviedo, Spain; manufsanjurjo@gmail.com (M.F.-S.); paolapintohh2@gmail.com (P.P.-H.); juan.cas.cas@gmail.com (J.C.-S.); cotoalmudena@uniovi.es (A.C.-V.); tomascristina@uniovi.es (C.T.-Z.); iglesiaseduardo@uniovi.es (E.I.-G.); 2Translational Interventions for Health (ITS) Group, Health Research Institute of the Principality of Asturias (ISPA), 33011 Oviedo, Spain; 3University Institute of Oncology (IUOPA), University of Oviedo, 33006 Oviedo, Spain; fernandezfjavier@uniovi.es (J.F.); cjvg@uniovi.es (C.J.V.); lombofelipe@uniovi.es (F.L.); 4Department of Functional Biology (Microbiology), University of Oviedo, 33006 Oviedo, Spain; 5Biotechnology of Nutraceuticals and Bioactive Compounds (BIONUC) Group, Health Research Institute of the Principality of Asturias (ISPA), 33011 Oviedo, Spain; 6Department of Biomedical Data Science, Geisel School of Medicine at Dartmouth, Hanover, NH 03755, USA; pablo.martinez-camblor@hitchcock.org; 7Faculty of Health Sciences, Universidad Autónoma de Chile, Providencia 7500912, Chile; 8Mitchelton-Scott, Via Campo di Maggio 35, 21020 Varese, Italy; manuelra@nutrimaxper.com; 9Research Centre for High Performance Sport, Catholic University of Murcia, 30107 Murcia, Spain; rortolano@ucam.edu; 10Basic-Clinical Research in Neurology Group, Health Research Institute of the Principality of Asturias (ISPA), 33011 Oviedo, Spain; 11Department of Microbiology and Biochemistry of Dairy Products, Dairy Research Institute of Asturias-Spanish Research Council (IPLA-CSIC), 33300 Villaviciosa, Spain; lorena.ruiz@ipla.csic.es (L.R.); amargolles@ipla.csic.es (A.M.); 12Functionality and Ecology of Beneficial Microbes (MicroHealth) Group, Health Research Institute of the Principality of Asturias (ISPA), 33011 Oviedo, Spain; 13Department of Morphology and Cell Biology (Anatomy), University of Oviedo, 33006 Oviedo, Spain

**Keywords:** metagenomics, probiotics, metabolomics, prebiotics, sports nutrition, exercise performance

## Abstract

Background: Regular exercise has been described to modify both the diversity and the relative abundance of certain bacterial taxa. To our knowledge, the effect of a cycling stage race, which entails extreme physiological and metabolic demands, on the gut microbiota composition and its metabolic activity has not been analysed. Objective: The aim of this cohort study was to analyse the dynamics of faecal microbiota composition and short-chain fatty acids (SCFAs) content of professional cyclists over a Grand Tour and their relationship with performance and dietary intake. Methods: 16 professional cyclists competing in La Vuelta 2019 were recruited. Faecal samples were collected at four time points: the day before the first stage (A); after 9 stages (B); after 15 stages (C); and on the last stage (D). Faecal microbiota populations and SCFA content were analysed using 16S rRNA sequencing and gas chromatography, respectively. A principal component analysis (PCA) followed by Generalised Estimating Equation (GEE) models were carried out to explore the dynamics of microbiota and SCFAs and their relationship with performance. Results: *Bifidobacteriaceae*, *Coriobacteriaceae*, *Erysipelotrichaceae*, and *Sutterellaceae* dynamics showed a strong final performance predictive value (r = 0.83, ranking, and r = 0.81, accumulated time). Positive correlations were observed between *Coriobacteriaceae* with acetate (r = 0.530) and isovalerate (r = 0.664) and between *Bifidobacteriaceae* with isobutyrate (r = 0.682). No relationship was observed between SCFAs and performance. The abundance of *Erysipelotrichaceae* at the beginning of La Vuelta was directly related to the previous intake of complex-carbohydrate-rich foods (r = 0.956), while during the competition, the abundance of *Bifidobacteriaceae* was negatively affected by the intake of simple carbohydrates from supplements (r = −0.650). Conclusions: An ecological perspective represents more realistically the relationship between gut microbiota composition and performance compared to single-taxon approaches. The composition and periodisation of diet and supplementation during a Grand Tour, particularly carbohydrates, could be designed to modulate gut microbiota composition to allow better performance.

## 1. Introduction

Gut microbiota composition influences the risk of highly prevalent pathologies, such as cardiovascular and metabolic diseases or cancer [1,2], although the exact mechanisms through which this effect takes place are diverse and yet not well defined [3].

Regular exercise has been described as a factor modifying both the diversity and the relative abundance of certain bacterial phyla or families [4,5], although the few studies that have been carried out on this show important methodological differences. The main divergence between studies lies in the species used (human, rat, mouse); the type of sample in which the microbiota is analysed (faeces and/or caecal content); the presence or absence of dietary control; the frequency, duration, and intensity of exercise interventions (acute exercise or training) [6,7,8]; and exercise type (endurance, resistance, concurrent training, etc.) [9].

Some authors have described that gut microbiota composition and metabolic activity directly influence physical performance [10,11,12]. This puts the focus on the use of sports supplements that could modulate the composition and/or metabolic activity of the gut microbiota in order to improve performance [13]. Although it has been described that gut microbiota composition can be modified with the use of different supplements (probiotics, prebiotics, proteins, antioxidants, branched-chain amino acids, caffeine, etc.) [14], the lack of solid information on how exercise modifies the composition of the gut microbiota, especially in high-level athletes during competition, makes this supplementation imprecise and potentially ineffective.

To our knowledge, the effect of a cycling stage race on the gut microbiota composition and metabolic activity of professional cyclists has not been analysed. Grand Tours involve repeated and continuous days of competition, which entail extreme physiological and metabolic demands and responses [15,16]. Therefore, studying the composition of the intestinal microbiota and the products of its metabolic activity in this context can help both to understand its potential modulatory role in the response to exercise and to optimise and personalise the use of supplements.

We hypothesise that performance in professional cyclists is correlated to changes in gut microbiota composition and is related to dietary intake and physiological stress factors during the three-week period of a Grand Tour. Thus, the aim of this study was to analyse the faecal microbiota composition and short-chain fatty acid (SCFA) content of professional cyclists over the three weeks of a Grand Tour, La Vuelta 2019, and its relationship with performance indicators, dietary intake, and supplement use.

## 2. Materials and Methods

### 2.1. Subjects

A convenience sample of 16 professional cyclists from two cycling teams among those competing in La Vuelta 2019 was recruited to participate in this study. Both teams were ranked in the top 10 of the 2019 UCI World Ranking, and both finished in the top 5 in the team classification at La Vuelta 2019. All the cyclists in those teams were contacted, accepted to participate, and signed an informed consent form. At the time of recruitment, and at least in the previous month, none of them was using any antibiotic or other pharmacological treatment that could interfere with the intestinal microbiota and that would have constituted a cause for exclusion. During the study, none of them was excluded due to the use of antibiotics. One of the participants suffered a crash that forced his withdrawal from the competition. Thus, the total number of subjects who completed the study was 15 cyclists (Figure 1).

Both the study design and the informed consent were reviewed and authorised by the Principality of Asturias Ethics Committee (Ref.: 238/19), and the research was performed in accordance with the Declaration of Helsinki.

### 2.2. Anthropometric Measurements

Body mass and height were measured using a medical scale with a measuring rod (Seca, model 704s; precision: 0.1 kg for weight and 0.1 cm for height; Hamburg, Germany) at the beginning and end of the competition. Body mass index (BMI) was then calculated as the Quetelet index, i.e., as body mass in kilograms divided by the square of the height in metres.

### 2.3. Collection of Faecal Samples

Faecal samples were collected at four time points: on the morning of the day before the first stage of La Vuelta 2019 (A); on the morning of the first rest day (after 9 stages) (B); on the morning of the second rest day (after 15 stages) (C); and on the morning of the last day of the race (after 20 stages) (D).

Sampling was performed according to established protocols to avoid cross-contamination using Faecal Nucleic Acid Collection and Preservation Tubes (Norgen Biotek Corp., Thorold, ON, Canada) and kept at −20 °C until analysis to preserve the nucleic acids and metabolites in perfect condition for subsequent analysis.

### 2.4. Analysis of Gut Microbiota Populations Using 16S rRNA Sequencing

From 200 mg of faecal samples, genomic DNA (gDNA) was extracted using the E.Z.N.A.^®^ DNA extraction kit (Omega Bio-Tek Ref. D4015-02, Norcross, GA, USA), obtaining 200 µL of gDNA. The gDNA samples were quantified using the BioPhotometer^®^ (Eppendorf, Hamburg, Germany), and their concentrations were diluted and standardised to 6 ng/µL.

These samples were used for PCR amplification using the Ion 16^TM^ Metagenomic kit (Thermo Fischer Scientific, Waltham, MA, USA). The PCR amplification products were used to create a genomic library using the Ion Plus Fragment Library kit AB library Builder^TM^ System (Cat. No. 4477597, Thermo Fischer Scientific, Waltham, MA, USA). Finally, the Ion Xpress^TM^ Barcode adapters 1–96 kit (Cat. No. 4474517, Thermo Fischer Scientific, Waltham, MA, USA) was added to each sample for subsequent sequencing.

Sequencing was performed using the ION PGM^TM^ Hi-Q^TM^ Sequencing kit (Cat. No. A25592, Thermo Fischer Scientific, Waltham, MA, USA) on the ION PGM^TM^ system. The chip used was the 318^TM^ v2 (Cat. No. 4484355, Thermo Fischer Scientific, Waltham, MA, USA). The total number of reads was always greater than 110,000 per sample.

The consensus table for each sequencing run was downloaded from the ION Reporter 5.6 software. This table includes the percentages for each taxonomic level and was used to compare frequencies between individuals. Taxonomic adscription down to the species level was performed using QIIME 2 (v.2017.6.0). The reference libraries used were the Curated MicroSEQ^TM^ 16S Reference Library v2013.1 and Greengenes v13.5. Those phyla, families, or species that did not reach 0.1% of relative abundance at any of the sampling points were eliminated for further analysis. From the sequencing information generated, the Shannon diversity index, also known as the richness and evenness index [17], was calculated using the QIIME 2 platform. All raw metagenomics data have been deposited in the NCBI SRA database (Accession number: PRJNA645285).

### 2.5. Short-Chain Fatty Acid Analysis

The SCFAs acetate, propionate, isobutyrate, butyrate, isovalerate, and valerate were determined in faeces through gas chromatography (GC), using a system composed of a 6890NGC injection module (Agilent Technologies Inc., Palo Alto, CA, USA) connected to a flame injection detector (FID) and a mass spectrometry (MS) 5973N detector (Agilent), as previously described. Briefly, cell-free supernatants (100 μL) from faecal homogenates were mixed with 450 μL methanol, 50 μL internal standard solution (2-ethylbutyric 1.05 mg/mL), and 50 μL 20% *v*/*v* formic acid. Following centrifugation, the supernatants of this mixture were used for SCFA quantification by GC as previously described [18].

### 2.6. Performance, Fatigue Perception, and Recovery

Three objective parameters were measured as indicators of performance: power output, expressed as the average power-to-weight ratio per stage (W/kg), and position in the overall individual ranking and accumulated time at each sampling point, obtained from the data of the official competition record (https://www.lavuelta.es/es/historia, accessed on 20 February 2024).

On the other hand, two subjective parameters were measured: the Rating of Perceived Exertion (RPE) at the end of La Vuelta, as an indicator of fatigue, and the Total Quality Recovery (TQR) scale, both at the beginning and at the end of the competition, as an indicator of recovery. For the analysis of the RPE, the Borg scale 6–20 [19] was used, being 6 “very very light” and 20 “very very hard”. Perception of recovery was evaluated using a 6–20 scale [20], with 6 being “very, very poor recovery” and 20 being “very, very good recovery”.

### 2.7. Dietary Assessment and Recording of Probiotic Supplement Use

Food intake and supplement use recording before and during La Vuelta was only provided by the medical staff of one of the teams (n = 8).

A retrospective analysis of dietary intake during the month preceding the start of La Vuelta was carried out using a validated food frequency questionnaire [21], which was modified based on a quantitative study on the food consumption of elite cyclists during training and competition [22] and included specific sports foods [23]. The food items added to the original questionnaire were: wholegrain cereals and derivatives (bread, pasta, rice, breakfast cereals, cookies, etc.), olive oil, other vegetable oils (sunflower, corn, canola, etc.), butter, margarine, and fermented dairy products other than yoghurt (kefir, koumiss, etc.). Moreover, items corresponding to milk and dairy products were itemised as: whole, semi-skimmed, and skimmed milk, and full-fat yoghurt, low-fat yoghurt, and fermented dairy products. Regarding sports foods, the use of carbohydrate (CHO) drinks, CHO/protein bars, and CHO/caffeinated gels was also included.

The same questionnaire was used to assess food intake during competition. The cyclists were asked to record their intake two days before the B sampling point. In agreement with the teams’ medical staff, the cyclists were asked to repeat the same dietary pattern two days before C and D sampling points to minimise the influence of this variable. The use and composition of probiotic supplements throughout the competition were also recorded for each cyclist by the medical staff, as was any antibiotic treatment.

### 2.8. Statistical Analysis

Pearson’s correlation coefficients and simple linear regression were used for studying the association between variables. Comparisons between groups were based on a paired Student’s T test or Wilcoxon signed-rank test (for TQR, body composition, and diet) and repeated measures ANOVA or Friedman test (alpha diversity index, phyla abundance, ranking position, accumulated time, and power output) as appropriate (depending on parametric or non-parametric datasets, respectively). A Principal Component Analysis (PCA) was performed at point A for both microbiota composition at the family level (the lowest taxonomic category for which the sequencing technique used allows the identification of 100% of the taxa) and SCFA. PCA had a two-fold objective: (i) a dimensionality reduction and (ii) identifying latent components of the gut microbiota and SCFAs. Using varimax rotation, we extract components explaining at least 50% of the total variance. The model was applied at the other sampling points (B–D). Finally, in order to consider the effects of repeated measures on the same subjects, Generalised Estimating Equation (GEE) models with an unstructured variance-covariance matrix structure were used for modelling the relationship between the performance indicators and the microbiota and SCFA components. Standard linear regression models were used for developing prediction models at the end of the race. A leave-one-out cross-validation procedure was used to reduce the overfitting in its accuracy evaluation. All reported *p*-values were two-sided, and those below 0.05 were considered statistically significant. R.4.2.3 (www.r-project.org, accessed on 20 February 2024) was used for statistical analysis.

## 3. Results

### 3.1. Gut Microbiota and SCFA Composition Varies throughout La Vuelta

We studied 15 professional cyclists at four different timepoints during La Vuelta 2019, whose characteristics are shown in Table 1. Considering all sampling points, the faecal 16S metagenomic dataset yielded a total of 8 phyla, 42 families, 45 genera, and 75 species (Appendix A, phyla; S2, families; and S3, genera and species). The sequencing technique used allowed the identification of 100% of the taxa from phylum to family levels. Shannon and Simpson indexes were calculated at family level, and no significant differences were observed between sampling points (A–D), being 3.3 ± 0.4, 3.1 ± 0.4, 3.1 ± 0.4, and 3.2 ± 0.3 (*p* = 0.405) and 0.81 ± 0.06, 0.81 ± 0.06, 0.79 ± 0.06, and 0.79 ± 0.05 (*p* = 0.433), respectively (Appendix A).

We observed changes in the abundance of certain phyla and families between some sampling points. Thus, the relative abundance of the phylum Actinobacteria increased significantly (*p* = 0.000331) from A (0.6 ± 1.6%) to B (3.0 ± 3.4%) (Figure 2). As the scenario was more complex at the family level, we decided to perform a PCA in order to understand gut microbiota dynamics throughout La Vuelta. This PCA was defined at point A and then applied at the other sampling points (B–D). Three principal components (PC1, PC2, and PC3) explained 50% of the variance in family abundance at A (Figure 3A). The dynamics of this model is shown in Figure 3B. The main families included in these PCs (the absolute value of the load larger than 0.5) and their specific responses are shown in Figure 3C for PC1, 2D and 2E for PC2, and 2F for PC3.

A PCA was also carried out following the same approach to analyse the dynamics of faecal SCFAs during La Vuelta (Figure 4A,B). One principal component (PC1) explained 70% of the variance at A, with all SCFAs participating with a positive score. The specific response of each SCFA is shown on Figure 4C.

To explore the relationship between faecal microbiota composition and SCFA concentration, GEE models were used. A positive relationship was observed between microbiota PC1 and isovaleric (*p* < 0.0001) and isobutyric (*p* = 0.043) acids. Moreover, microbiota PC2 and PC3 showed a negative relationship with SCFAs. Specifically, PC2 with propionic acid (*p* = 0.048), isobutyric (*p* < 0.0001), butyric (*p* < 0.0001), isovaleric (*p* = 0.002), and valeric (*p* < 0.0001) acids; and PC3 with butyric (*p* = 0.034) and valeric (*p* = 0.011) acids. A forest plot with these relationships is shown in Figure 4D. Considering that an increase in SCFA faecal content was observed at sampling point B (Figure 4B), a correlation analysis was performed at this sampling point between specific SCFA concentrations, and the abundance of those families related to performance according to GEE models, as well as some of its genera and species. Positive correlations were observed at this sampling point for *Coriobacteriaceae* and its species *Collinsella aerofaciens* with acetic (Pearson correlation coefficient of 0.530, 95% CI: 0.024–0.820, *p* = 0.042; and 0.698, 95% CI: 0.290–0.892, *p* = 0.004) and isovaleric acids (Pearson correlation coefficient of 0.664, 95% CI: 0.230–0.878, *p* = 0.007; and 0.549, 95% CI: 0.052–0.829, *p* = 0.034) and for *Bifidobacteriaceae* and its genus *Bifidobacterium* with isobutyric acid (Pearson correlation coefficient of 0.682, 95% CI: 0.262–0.885, *p* = 0.005; and 0.682, 95% CI: 0.262–0.885, *p* = 0.005) (Appendix A). No correlation was observed between *Sutterellaceae* abundance and SCFA concentration at this sampling point.

Interestingly, sampling point B was also relevant for performance. A strong, significant positive correlation was observed between the classification at points B and D (Pearson correlation coefficient of 0.951, 95% CI: 0.855–0.984; *p* < 0.0001) (Figure 5A). Furthermore, we observed that those cyclists performing a higher average power-to-weight ratio per stage from A to B obtained better results in terms of better final ranking (Pearson correlation coefficient of −0.781, 95% CI: (−0.927)–(−0.427); *p* = 0.000981) (Figure 5B) and lower accumulated time at the end of La Vuelta (Pearson correlation coefficient of −0.818, 95% CI: (−0.941)–(−0.508); *p* = 0.000347) (Figure 5C). This correlation was not observed from B to C or from C to D, although we found it between the average power-to-weight ratio per stage throughout La Vuelta and both final ranking (Pearson correlation coefficient of −0.627, 95% CI: (−0.875)–(−0.116); *p* = 0.022) (Figure 5D) and final accumulated time (Pearson correlation coefficient of −0.642, 95% CI: (−0.881)–(−0.141); *p* = 0.018) (Figure 5E).

### 3.2. Gut Microbiota Dynamics during La Vuelta Predicts Performance

In addition to analysing changes in gut microbiota composition and SCFA concentration throughout La Vuelta, we recorded performance, expressed as the position in the overall individual ranking and as accumulated time (Table 2). Based on a linear regression model, we explored the relationship between gut microbiota PCA models and accumulated time and ranking at the end of La Vuelta (Figure 6). Using PC1, PC2, and PC3 together, a strong performance predictive value was observed, both for accumulated time (Pearson correlation coefficient of 0.83, *p* = 0.0004348, Figure 6A) and for ranking (Pearson correlation coefficient of 0.81, *p* = 0.0006532, Figure 6B). Focusing specifically on each PC, we observed that those cyclists with high levels of PC2 at A, high levels of PC1 and PC2 at B, and low levels of PC2 and high levels of PC3 at C performed better, obtaining a higher final ranking, i.e., less accumulated time. Deepening this analysis, the specific representative families within each PC that follow these trends are *Bifidobacteriaceae* and *Coriobacteriaceae* for PC1, *Erysipelotrichaceae* for PC2, and *Sutterellaceae* for PC3 (Figure 3C,E,F).

Interestingly, the models provided no relationship between SCFAs and performance. Furthermore, no relationship was observed between the dynamics of microbiota and SCFA with other performance, recovery, and fatigue parameters, such as the average power-to-weight ratio per stage, TQR or RPE.

### 3.3. Dietary Intake Modifies Microbiota Composition to Modulate Performance

Considering that both microbiota composition and physical performance are modulated by diet, food intake, and sport supplement use, including probiotics, in the month prior to La Vuelta and during the competition were recorded by means of a food frequency questionnaire. The results are shown in Table 3. A significant increase was observed during La Vuelta in the frequency of consumption of food groups rich in refined CHO, such as pasta (percent variation = 350 ± 382%), rice (177 ± 111%), bread (114 ± 210%), or soft drinks (136 ± 95%), although the intake of these food groups was already high previously. Similarly, the consumption of high-CHO sport supplements increased significantly during competition (CHO drinks: 442 ± 677%; gels: 533 ± 47%; energy bars: 288 ± 214%; sport snacks: 167 ± 145%). The main probiotic food consumed was yoghurt, which includes members of the genus *Lactobacillus* and the species *Streptococcus thermophilus.* The athletes studied did not consume probiotic supplements in the weeks prior to La Vuelta, although all used these supplements daily during the competition. The commercial products used included combinations of the species *Bifidobacterium lactis*, *Bifodobacterium bifidum*, *Lactobacillus paracasei*, and *Lactobacillus acidophilus.*

The influence of dietary intake during the month preceding the start of La Vuelta on the abundance at sampling point A of those families more closely related to performance was analysed. A significant positive correlation was observed between *Erysipelotrichaceae* relative abundance and potato intake (Pearson correlation coefficient of 0.956, 95% CI: 0.771–0.992; *p* = 0.000203) (Figure 7A). According to the performance prediction analysis presented above, high levels of PC2, where *Erysipelotrichaceae* participates, in A are related to a lower accumulated time at the end of La Vuelta and therefore, better performance.

We also studied this relationship during the competition. Those cyclists with the lower frequency of consumption of CHO-rich supplements (CHO drinks, gels, and sport snacks, taken together) during La Vuelta showed a tendency towards a higher relative abundance of the family *Bifidobacteriaceae* at sample point B (Pearson correlation coefficient of −0.652, 95% CI: −0.930–0.097; *p* = 0.080, Figure 7B), where the abundance of this taxon increases considerable regarding A (Figure 3C). Moreover, although the consumption of high-CHO sport supplements increased significantly during competition, as mentioned before, those cyclists with the lower increase in their consumption also showed a tendency towards a higher abundance of *Bifidobacteriaceae* family at B (Pearson correlation coefficient of −0.650, 95% CI: −0.929–0.101; *p* = 0.081, Figure 7C). It is worthy to mention that, according to the performance prediction analysis presented above, high levels of PC1, where *Bifidobacteriaceae* participates, in B are related to a lower accumulated time at the end of La Vuelta and therefore, better performance.

The species consumed in both probiotic foods and supplements belonged to the *Bifidobacteriaceae* and *Lactobacillaceae* families. The former is important in PC1 (Figure 3A) and related to performance, as mentioned above. The latter was not relevant in any of the PCs (Figure 3A) and, therefore, did not intervene in the prediction of performance.

## 4. Discussion

This is the first study to analyse changes in faecal microbiota composition and SCFA concentration during a Grand Tour cycling race. We have also been able to identify those taxa whose timing and magnitude of change during competition show a very close relationship with performance and could contribute to its prediction. Accordingly, this study provides unique information on bacterial dynamics and metabolic activity in response to stress imposed by the accumulation of exercise loads during three weeks of competition. Our results show that the dynamics of the microbiota during La Vuelta involve many taxa, with diverse metabolic activity. Those dynamics result from the demands imposed by exercise and are related to dietary intake, both prior to and during La Vuelta.

The mechanisms by which gut microbiota may influence exercise performance have not yet been fully elucidated, although it appears that microbial SCFA production may play a role in muscle energy metabolism, both by being used directly and by affecting the availability of other substrates [10,24,25]. In fact, well-known families involved in SCFA production were relevant in the microbiota dynamics during La Vuelta [26], such as acetate producers such as *Bifidobacteriaceae* and *Coriobacteriaceae* and diverse propionate and butyrate producers, such as *Oscillospiraceae* and *Veillonellaceae,* from PC1; *Bacteroidaceae*, *Prevotellaceae,* and *Erysipelotrichaceae,* from PC2; and *Lachnospiraceae*, *Ruminococcaceae*, *Porphyromonadaceae,* and *Sutterellaceae*, from PC3 [26].

Our results show that several of these SCFA-producing families allow performance prediction according to the GEE models, such as *Coriobacteriaceae* and *Bifidobacteriaceae*, from PC1, *Erysipelotrichaceae* from PC2, and *Sutterellaceae* from PC3. These families have previously been described as associated with improved physical performance and exercise-related health status in mouse and human studies [27,28]. Thus, *Coriobacteriaceae* abundance has been described as a biomarker linking physical exercise with improved health status, probably via diverse catabolic pathways, such as secondary bile acids and aldosterone 18-glucuronide metabolism, involved in sodium homeostasis [29]. The genus *Bifidobacterium,* the main member of the family *Bifidobacteriaceae,* has been associated to improved sport performance and lower plasma levels of parameters associated with fatigue, such as lactate, ammonia, creatine kinase, and inflammatory biomarkers such as pro-inflammatory cytokines, without changes in SCFA production [30,31]. In addition, *Erysipelotrichaceae* was found to be associated with VO_2_peak due to its ability to increase butyrate production [32], while *Sutterella*, the main genus belonging to the family *Sutterellaceae*, has been positively associated with this parameter through its effect on antioxidant defences [33]. Therefore, despite being well-known SCFA producers, the mechanism by which these families are related to performance improvements is not only mediated by their ability to produce SCFAs but also other metabolites. Interestingly, we have not found a relationship between faecal SCFA content and athletic performance, probably because their faecal content only accounts for about 5% of total SCFA production [34], which represents the balance between production, direct uptake by intestinal epithelial cells, and diffusion from the lumen into the portal bloodstream [35]. Once these colonic SCFAs are absorbed into the portal vein (mainly acetate and propionate) and distributed to the liver and other distant organs, they continue performing important functions. For example, at the muscular level, they contribute to the correct functioning of energy metabolism processes, facilitating the beta-oxidation of fats and the possible recovery of muscle glycogen [36]. Therefore, we cannot rule out the potential role of SCFAs on performance during La Vuelta, despite the lack of correlation observed with their faecal content. Thus, some members of the *Veillonellaceae* and *Lachnospiraceae* families, well-known SCFA producers, have been reported to utilise lactate to synthetise propionate [37], which highlights their potential value in the context of exercise performance [11,38]. In fact, Scheiman et al. [11] demonstrated in animal models that serum lactate can reach the intestinal lumen by crossing the intestinal epithelial barrier and that intrarectal infusion of propionate improves performance. In addition, *Lachnospiraceae* is a major producer of butyrate. A positive correlation between butyrate production and VO_2_peak has been described [32]. However, the response dynamics of these families are inconsistent with the performance prediction model obtained in our study. Their role as performance enhancers, if any, does not seem to be related to their abundance but maybe to their metabolic interaction with other taxa, probably as part of the so-called cross-feeding phenomenon [39].

Interestingly, although the abundance of *Veillonellaceae* was not related to diet at any sampling point, the relative abundance of *Lachnospiraceae* at the beginning of La Vuelta was directly related to the intake of high-fibre foods (whole-grain foods, vegetables, fruits, potatoes, and legumes, taken together) in the preceding weeks (Pearson correlation coefficient of 0.919, 95% CI: 0.608–0.985; *p* = 0.001, Appendix A), as previously described in the non-athletic population [40]. However, this relationship was not observed during competition at any point, even though consumption of fibre-rich foods did not change significantly (46.9 ± 23.6 vs. 53.2 ± 24.3 servings per week; *p* = 0.401). This suggests that, although dietary fibre has a clear modulatory effect on the abundance of the *Lachnospiraceae* family, this is less important in the presence of repeated competitive exercise loads when fibre intake is maintained. Alternatively, this absence of correlation during the competition could be due to other variables, such as an increased total caloric intake or high glycemic index foods, or the accumulated physiological stress along the race. It would remain to be elucidated what would happen with respect to the relationship of this family to performance if the intake of fibre-rich foods were modified in a targeted manner at specific times during competition. Similarly, the relative abundance of *Erysipelotrichaceae* at the beginning of La Vuelta was directly related to the intake of complex CHO food sources, although this relationship was not observed during competition. According to our performance prediction model, the higher the relative abundance of this family at A, the better for performance. Therefore, in this case, the modulatory role of the pre-competition diet has an impact on performance. Regarding the effect of food intake during competition on the abundance of other bacterial families related to performance, we have found that *Bifidobacteriaceae* is unable to progress in the gut during competition under high intakes of fast digesting CHO from supplements (glucose, fructose, glucose polymers) or when drastically increasing their consumption habits of these types of supplements. This last idea is related to the concept of “gut training” for athletes proposed by Jeukendrup [41], according to which the adaptations that allow greater intestinal and metabolic utilisation of CHO are greater when a diet high in CHO is regularly followed. Our data show that the dynamics of microbiota composition during competition are sensitive to dietary changes that have an impact on sports performance, suggesting that this may be another element to take into account within this “gut training” paradigm. These results are relevant since they highlight the potential of diet, particularly simple and complex CHO, to mediate or modify performance through the modulation of gut microbiota composition before and during competition.

Interestingly, although our results focus on the response to three weeks of competition in professional cyclists, some authors have analysed the composition of the intestinal microbiota in amateur cyclists, as well as its relationship with CHO intake and metabolism. Thus, Petersen et al. 2017 [42] recruited a group of professional and amateur cyclists. Using mWGS sequencing data, they showed that the gut microbiomes of all cyclists analysed together fell into three taxonomic groups: high *Prevotella*, high *Bacteroides*, or a mixture of many genera, including *Bacteroides*, *Prevotella*, *Eubacterium*, *Ruminococcus*, and *Akkermansia*. While no significant correlations were found between professional and amateur cyclists, the abundance of the genus *Prevotella* correlated significantly with training time per week, although not all amateur cyclists trained less time than professional cyclists. In addition, metatranscriptome analysis revealed that *Methanobrevibacter smithii* transcriptional activity was higher in professional cyclists. Both taxa were functionally related to energy and carbohydrate metabolism pathways. For their part, Wiacek et al., 2023 [43] studied how the abundance of four species present in the faecal microbiota (*Faecalibacterium prausnitzi*, *Akkermansia muciniphila*, *Bifidobacterium* spp., and *Bacteroides* spp.) changed at two points throughout the season in amateur cyclists versus sedentary volunteers collected at a single point. They found no differences in the abundance of these species between amateur and sedentary cyclists. Although the authors note that amateur cyclists increase their consumption of single CHO throughout the season, the abundance of the species analysed did not change. The reason for this discrepancy from our results could be that the physiological stress of a grand tour cycling competition in professional cyclists, as well as the training load, is much higher than the demands of amateur cyclists. Furthermore, to our knowledge, there is no study in which changes in gut microbiota composition during a competition in amateur cyclists have been analysed. In contrast, the above studies have only analysed the gut microbiota composition and metabolic activity at rest in the trained state.

Future strategies to enhance the performance of high-level athletes could include the selective growth of specific targeted bacterial taxa using *à la carte* prebiotics instead of the current ineffective probiotic supplementation, which usually cannot maintain stable populations in the gut ecosystem. In fact, we have been unable to detect any of the strains included in the probiotic supplements or in the probiotic foods used. Therefore, although the cyclists were taking probiotic supplements containing bifidobacteria during competition, the relationship between bifidobacteria abundance and performance appears to be modulated by diet in a way that is apparently more relevant to performance than the use of probiotic supplements.

## 5. Conclusions

In conclusion, the relationship of gut microbiota with sports performance is complex and not determined by single taxa or single metabolites, not even SCFAs, as suggested by certain authors [11,44]. An ecological perspective seems to represent a more realistic approach to the relationship between the composition and the metabolic activity of gut microbiota and performance. Under this paradigm, the composition and periodisation of diet and supplementation, if necessary, during a Grand Tour should be designed, not only from a physiological, energetic, or metabolic perspective, but also to modulate the composition of the gut microbiota towards that which could allow better performance at each moment of the competition, depending on the accumulated loads.

Some limitations can be identified in this study. Firstly, the observed relationship between gut microbiota composition and performance may be affected by other factors that simultaneously influence both parameters, mainly dietary intake. Although the frequency of food and sports supplement consumption before and during the competition was recorded in this study, it was only analysed in eight of the cyclists due to formal limitations from the professional teams. Furthermore, the methodology employed did not allow for an accurate measurement of energy or nutritional intake, so the associations found are not causally related to nutritional factors. Secondly, other uncontrolled factors, such as team strategies and individual roles (gregario, leader, sprinter, etc.), may also influence the relationship between microbiota composition and performance. It should be noted that these different roles do not necessarily mean differences in the effort undertaken, the energy expenditure during the stage, or even the final ranking position. This makes it complex to define, from a biological perspective, what is the highest or most successful performance in this sport. Thus, working for the team leader often leads to some riders not achieving a good ranking despite putting in a great effort. It can also happen that the leader of a team does not achieve a good final ranking, as was the case for the leader of one of the teams included in our study, outside the top 15 in the final ranking. Therefore, we measured not only the ranking or the accumulated time for each cyclist but also the accumulated power output, although this variable did not show such a strong relationship with the composition of the gut microbiota, probably because, as we mentioned, the effort exerted does not always have a relationship with the final ranking. Thirdly, the fact that blood samples were not available during the race is also a limitation to further study the modulatory mechanisms of the microbiota on performance, in particular to understand the functional role of SCFAs through the study of their presence in the bloodstream and their distribution to other tissues. Finally, it would be interesting to know if a single stage acutely modifies gut microbiota composition, if this change differs between types of stages (mountain, time-trial, etc.), and if gut microbiota composition varies throughout a season, between the different training periods.

## Figures and Tables

**Figure 1 nutrients-16-00661-f001:**
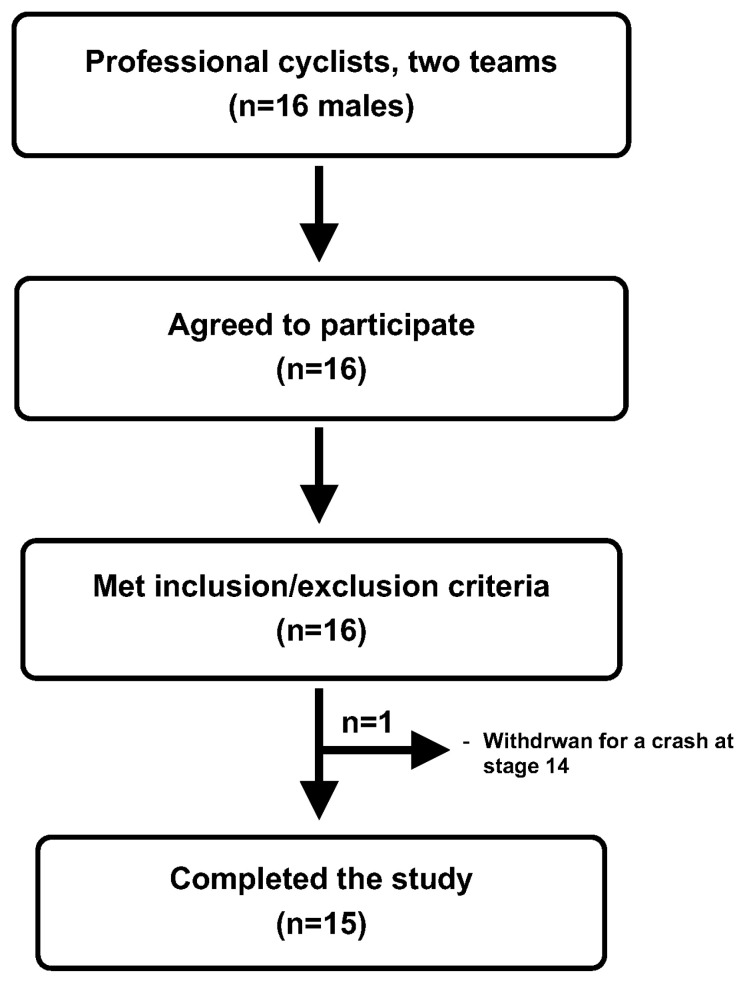
Schematic overview of participant recruitment.

**Figure 2 nutrients-16-00661-f002:**
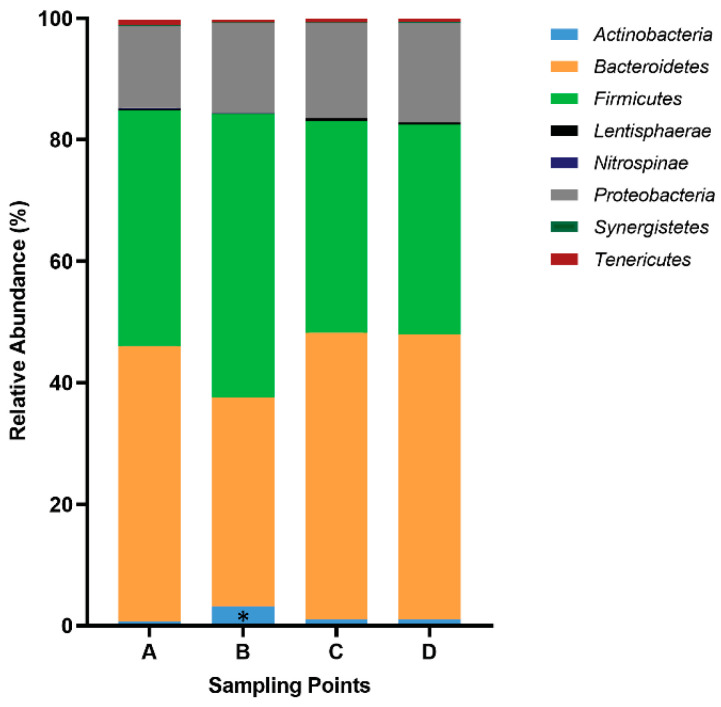
Relative abundance of phyla during La Vuelta. A: sampling point immediately before the first stage; B: sampling point after 9 stages; C: sampling point after 16 stages; D: sampling point after 20 stages. *: statistically significant difference between *Actinobacteria* A vs. B (*p* = 0.000331).

**Figure 3 nutrients-16-00661-f003:**
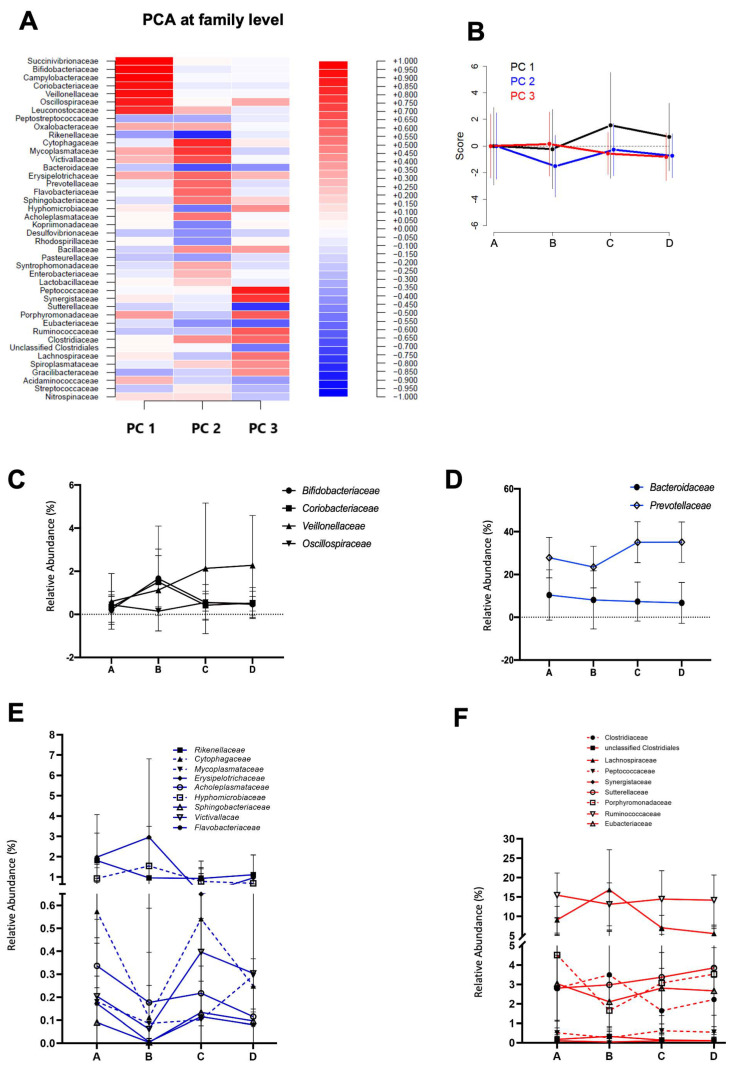
Microbiota dynamics during La Vuelta. (**A**). Heatmap of the principal component analysis (PCA) at the family level, including the coefficients for each family in the different PCs identified: Principal Component 1, PC1; Principal Component 2; PC2; Principal Component 3, PC3). (**B**). Dynamics of the PCs at the different sampling points, A–D. It shows the mean and standard deviation of the score obtained by applying the model at each sampling point. (**C**). Relative abundance of the families with coefficients < −0.5 or >0.5 in the PC1 at the different sampling points, A–D. (**D**). Relative abundance at the different sampling points, A–D, of the families with coefficients < −0.5 or >0.5 in the PC2 and relative abundances > 5%. (**E**). Relative abundance at the different sampling points, A–D, of the families with coefficients < −0.5 or >0.5 in the PC2 and relative abundances < 5%. (**F**). Relative abundance of the families with coefficients < −0.5 or >0.5 in the PC3 at the different sampling points, A–D. A: sampling point immediately before the first stage; B: sampling point after 9 stages; C: sampling point after 16 stages; D: sampling point after 20 stages.

**Figure 4 nutrients-16-00661-f004:**
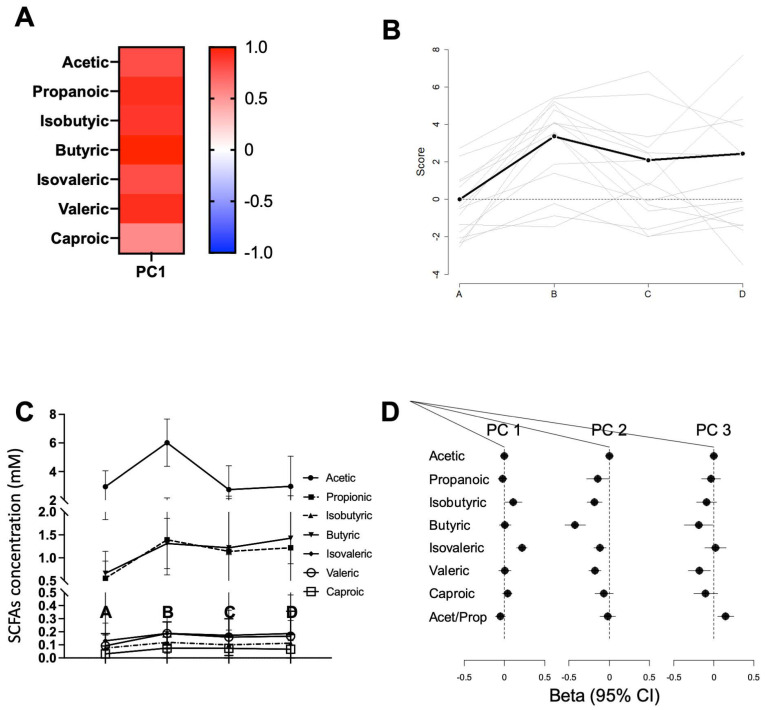
Short-chain fatty acid (SCFA) dynamics during La Vuelta. (**A**): Heatmap of the principal component analysis (PCA), including the coefficients for each SCFA in the PC identified. (**B**). Dynamics of the PC at the different sampling points, A–D. It shows the mean and standard deviation of the score obtained by applying the model at each sampling point. (**C**). Relative abundance of the SCFAs at the different sampling points, A–D. (**D**). Forest plot showing the global relationship between the principal components identified for microbiota (Principal Component 1, PC1; Principal Component 2, PC2; Principal Component 3, PC3) and SCFAs. The dots represent the estimated effect (beta), and the lines represent the 95% confidence interval. A: sampling point immediately before the first stage; B: sampling point after 9 stages; C: sampling point after 16 stages; D: sampling point after 20 stages.

**Figure 5 nutrients-16-00661-f005:**
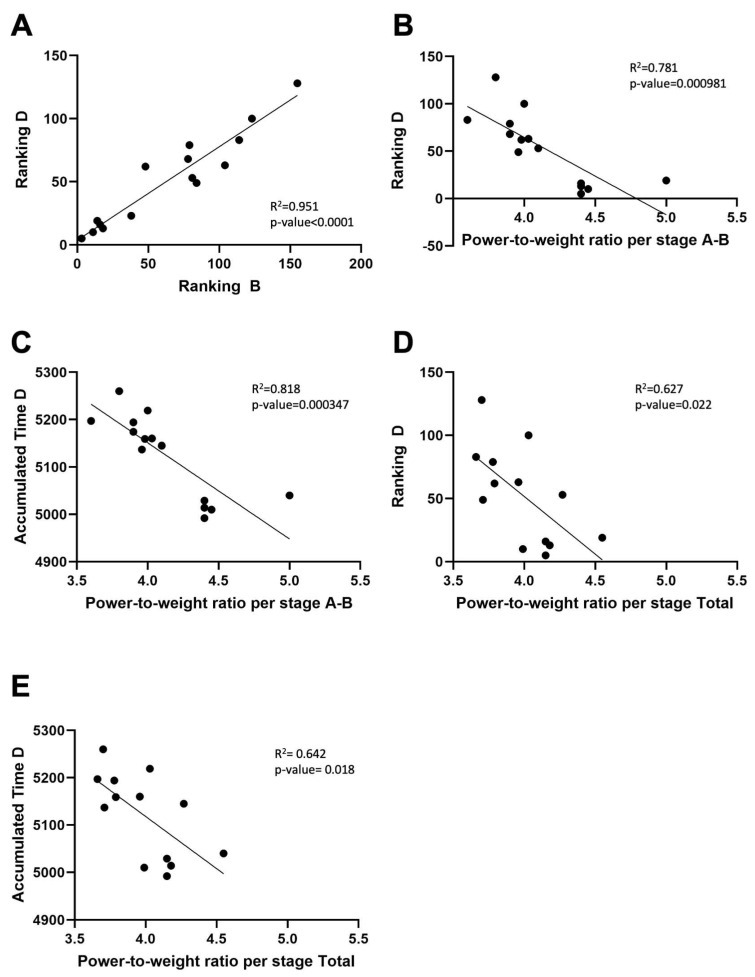
Performance parameters during La Vuelta. (**A**): Correlation analysis between ranking at points B and D. (**B**). Correlation analysis between final ranking (timepoint D) and power-to-weight ratio per stage between timepoints A and B. (**C**). Correlation analysis between final accumulated time and power-to-weight ratio per stage between timepoints A and B. (**D**). Correlation analysis between final ranking (timepoint D) and power-to-weight ratio per stage between timepoints A and D. (**E**). Correlation analysis between final accumulated time and power-to-weight ratio per stage between timepoints A and D. A: sampling point immediately before the first stage; B: sampling point after 9 stages; C: sampling point after 16 stages; D: sampling point after 20 stages.

**Figure 6 nutrients-16-00661-f006:**
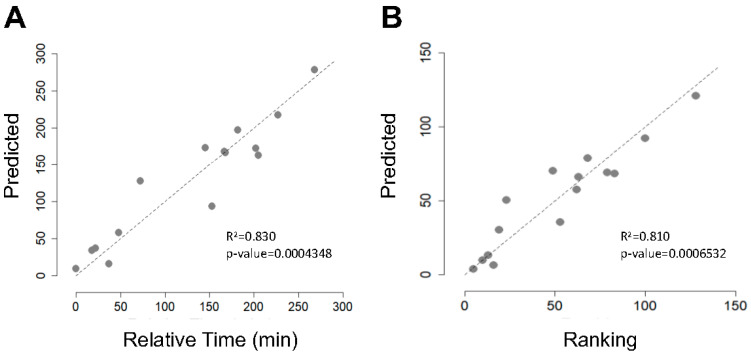
Performance prediction based on microbiota components. (**A**): Correlation analysis between predicted final accumulated time based on GEE models analysis and relative time, measured as the difference in the final accumulated time of each cyclist with respect to that of the best classified of those included in the study. (**B**): Correlation analysis between predicted final ranking based on GEE model analysis and final ranking.

**Figure 7 nutrients-16-00661-f007:**
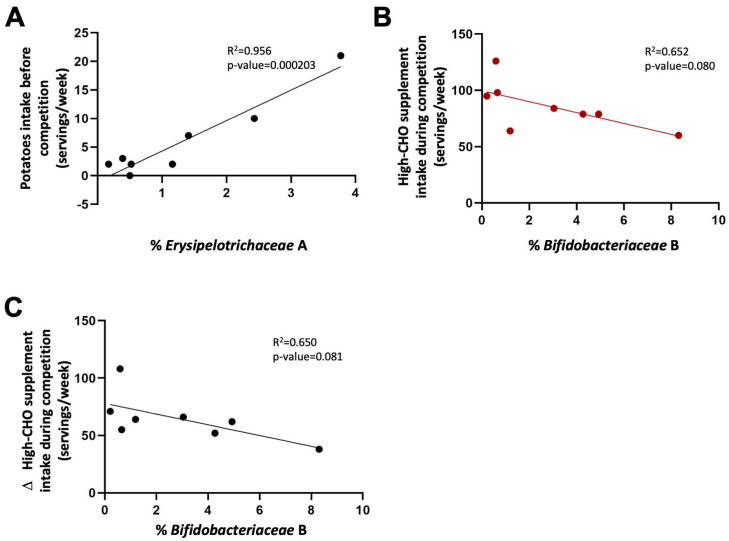
Modulatory effect of diet on microbiota composition before and during La Vuelta. (**A**): Correlation between *Erysipelotrichaceae* abundance at timepoint A and potato intake before La Vuelta. (**B**): Correlation between consumption of supplements high in simple CHO during La Vuelta and *Bifidobacteriaceae* abundance at timepoint B. (**C**): Correlation between *Bifidobacteriaceae* abundance at timepoint B and the difference in supplements high in simple CHO consumption during-before La Vuelta.

**Table 1 nutrients-16-00661-t001:** Physical characteristics of the subjects (n = 15).

Characteristics	Mean ± SD
Age (years)	30.2 ± 3.4
Height (cm)	178.4 ± 7.3
Initial weight (kg)	66.8 ± 5.0
Final weight (kg)	66.7 ± 5.0
Initial BMI (kg/m^2^)	21.0 ± 0.8
Final BMI (kg/m^2^)	21.0 ± 0.9

BMI: body mass index. SD: standard deviation.

**Table 2 nutrients-16-00661-t002:** Fatigue and recovery perception and performance parameters at different sampling points throughout competition (n = 15).

Parameter	Mean ± SD
TQR, A	17.7 ± 1.7
TQR, D	12.7 ± 7.3 *
RPE, D	16.3 ± 1.5
Classification position, B	67 ± 47 †
Classification position, C	56 ± 40
Classification position, D	51 ± 37
Accumulated time (min), B	2172.2 ± 38.2
Accumulated time (min), C	3829.3 ± 63.7
Accumulated time (min), D	5119.5 ± 86.7
Average power-to-weight ratio per stage (W·kg^−1^), A–B	4.2 ± 0.4 ‡
Average power-to-weight ratio per stage (W·kg^−1^), B–C	3.9 ± 0.3
Average power-to-weight ratio per stage (W·kg^−1^), C–D	3.8 ± 0.4
Average power-to-weight ratio per stage (W·kg^−1^), Total	4.0 ± 0.3

SD: standard deviation. A: sampling point before the first stage; B: sampling point after 9 stages; C: sampling point after 16 stages; D: sampling point after 20 stages. *: Statistically significant difference between TQR A vs. D (*p* = 0.000744); †: Statistically significant difference between classification position B vs. D (*p* = 0.017); ‡: Statistically significant difference between average power-to-weight ratio per stage A–B vs. B–C (*p* = 0.036) and C–D (*p* = 0.006).

**Table 3 nutrients-16-00661-t003:** Frequency of consumption (portions per week) of foods and sports supplements before and during La Vuelta 2019 (n = 8).

Food Item	Before Competition (Portions per Week)			During Competition (Portions per Week)			*p*-Value
* **Supplements** *							
Carbohydrate drinks	3.2	±	3.0	18.5	±	14.5	**0.012**
Gels	0.8	±	1.5	15.9	±	6.3	**0.011**
Energy bars	4.4	±	4.1	18.1	±	8.5	**0.018**
Sport snacks	12.8	±	11.1	33.1	±	12.6	**0.028**
Protein bars	1.1	±	1.6	1.9	±	2.8	0.285
*Foods*							
Breakfast cereals	4.6	±	7.2	10.0	±	11.5	**0.027**
Wholegrain breakfast cereals	3.1	±	3.4	4.4	±	5.2	0.109
Bread	11.1	±	9.1	18.4	±	10.9	**0.046**
Wholegrain bread	2.0	±	3.2	5.2	±	6.2	0.102
Rice	3.9	±	3.4	9.0	±	4.5	**0.011**
Brown rice	0.4	±	1.1	0.0	±	0.0	0.317
Pasta	2.5	±	1.8	7.6	±	2.7	**0.012**
Wholegrain pasta	0.2	±	0.7	0.0	±	0.0	0.317
Biscuits	2.5	±	4.2	8.1	±	6.2	**0.016**
Pastry products	3.4	±	7.3	4.9	±	7.4	0.109
Chocolate	3.6	±	6.9	4.8	±	12.3	1.000
Fruits	12.5	±	11.0	11.4	±	7.5	0.500
Fresh fruit juices	1.1	±	2.5	2.5	±	2.9	0.141
Commercial fruit juices	0.4	±	1.1	0.0	±	0.0	0.317
Leafy vegetables	8.0	±	4.7	9.1	±	4.9	0.396
Other vegetables	8.8	±	3.6	9.4	±	4.2	0.705
Potatoes	5.9	±	6.9	5.8	±	5.3	0.344
Legumes	3.9	±	4.8	5.0	±	6.6	0.343
Nuts	2.1	±	2.4	3.0	±	3.3	0.414
Whole milk	3.4	±	3.2	3.6	±	3.2	0.414
Semi-skimmed milk	1.8	±	3.2	2.6	±	5.2	0.317
Skimmed milk	0.9	±	2.5	0.9	±	2.5	1.000
Yoghurt	13.0	±	13.1	10.8	±	5.9	0.865
Other fermented milks	0.9	±	2.5	0.8	±	2.1	0.317
Fresh cheese	3.2	±	3.3	5.1	±	4.7	0.102
Mature cheese	1.1	±	2.5	2.0	±	4.9	0.317
Dairy desserts	0.5	±	0.9	1.8	±	2.8	0.180
Ice cream	1.0	±	2.4	0.8	±	1.2	0.655
Lean meat	4.8	±	1.7	5.1	±	1.6	0.180
Red meat	2.4	±	2.3	2.8	±	2.4	0.102
Meat products	0.4	±	0.5	0.2	±	0.5	0.317
Lean fish	2.1	±	3.1	2.6	±	3.0	0.180
Oily fish	3.4	±	2.4	3.2	±	2.6	1.000
Shellfish	0.4	±	0.7	0.2	±	0.7	0.317
Eggs	13.5	±	8.2	13.5	±	5.5	0.892
Olive oil	9.8	±	8.2	10.9	±	10.2	0.180
Other vegetable oils	0.0	±	0.0	0.0	±	0.0	1.000
Butter	7.1	±	6.1	8.1	±	8.3	0.343
Margarine	0.0	±	0.0	0.0	±	0.0	1.000
Salty snacks	1.2	±	2.4	0.9	±	2.5	0.180
Precooked food	0.0	±	0.0	0.0	±	0.0	1.000
Soft drinks	1.4	±	2.1	5.2	±	5.2	**0.043**
Diet drinks	0.9	±	2.5	0.0	±	0.0	0.317
Wine	1.2	±	1.6	1.6	±	1.4	0.180
Beer	1.6	±	3.5	0.8	±	1.5	0.180
Distilled alcoholic beverages	0.0	±	0.0	0.0	±	0.0	1.000

Data are presented as mean ± standard deviation. *p*-values in bold indicate statistically significant differences (*p* < 0.05) between before and during competition.

## Data Availability

All raw metagenomics data have been deposited in the NCBI SRA database (Accession number: PRJNA645285).

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
