# Peer review of "Dynamics of Gut Microbiota and Short-Chain Fatty Acids during a Cycling Grand Tour Are Related to Exercise Performance and Modulated by Dietary Intake"

_nutrients, 2024, doi:10.3390/nu16050661_

Round 1

Reviewer 1 Report

Comments and Suggestions for Authors

In the manuscript entitled “Dynamics of gut microbiota and short-chain fatty acids during a cycling Grand Tour are related to exercise performance and modulated by dietary intake” the authors aim to elucidate whether exercise can modify the composition of gut microbiota and content of short-chain fatty acids in high-level athletes during competition analyzing fecal microbiota in professional cyclers participating to La Vuelta 2019.

General comments

The manuscript provides an incremental knowledge towards the possibility to modify performance in elite professional cyclists during a major cycling stage race competition through the modulation of the gut microbiota optimizing the composition and periodization of diet and supplement.

Access to multiple samples during the development of La Vuelta 2019 gives the authors an unprecedented opportunity to analyze gut microbiota.

While the topic is interesting, the current manuscript is highly descriptive and no further insight on underlying mechanism(s) by which microbiota might impact cycler performance is provided.

As discusses by the authors, no conclusive role on SCFAs can be determined on the basis of the current study because their fecal content only accounts for about 5% of total SCFA production.

Minor limitation:

Line 101: “At the time of recruitment, and at least in the previous month, none of them was using any antibiotic or other pharmacological treatment that could interfere with the intestinal microbiota and that would have constituted a cause for exclusion”. What about pharmacological treatment during the 3 weeks of competition?

Line 215: “Supplementary Tables S1, phyla; S2, families; and S3, genera and species”. Clearly label supplemental file. Check decimal separator in tables. Why values from sample 2 are missing at B time-point?

Line 224: “Were observed changes in the abundance of certain phyla and families between sampling points”. Between some sampling points.

Line 492: “conclusion, the relationship of gut microbiota with sports performance is complex and not determined by single taxa or single metabolites, not even SCFAs, as suggested by certain authors”. As stated in the discussion (line 132 “we have not found a relationship between fecal SCFA content and athletic performance, probably because their fecal 433 content only accounts for about 5% of total SCFA production” line 441 “we cannot rule out the potential role of SCFAs on performance during La  Vuelta, despite the lack of correlation observed with their fecal content”) the no conclusive role or SCFAs can be determined on the basis of the current study.

Can authors discuss the major possible differences between elite professional bikers and possible controls (amateur road cyclists?).

Reviewer 2 Report

Comments and Suggestions for Authors

This cohort study was to analyse the dynamics of faecal microbiota composition and short-chain fatty acids (SCFAs) content of professional cyclists over a Grand Tour, and their relationship with performance and dietary intake. I have few comments:

Abstract:

To add more numbers.

Introduction:

-What is hypothesis of study?

Methods:

-What is registration in clinical trials?

-What is sample size calculus?

Results:

-Figure 3 is unable to read. Please, to increase the quality.

-Figure 5, please to add the line in the correlations.

-Table 3, could be used as supplementary file.

Discussion:

-This is great. But, please to add a discussion with different type of exercise as well as intensity of exercise. Its influency in microbiota.

Reviewer 3 Report

Comments and Suggestions for Authors

NUTR 2844812 Dynamics of gut microbiota and SCFA during a cycling Grand Tour are related to exercise performance and modulated by dietary intake

This manuscript describes an interesting analysis of the gut microbiota and faecal SCFA concentrations measured at different timepoints in 15 riders in the Vuelta. The samples are unique and there is a lot of interesting data. The data presentation is not optimal and there is a degree of overinterpretation of the results. My detailed comments are below.

Comments

1.     Methods, sampling times: at the beginning of the race, the participants of course likely had already trained a lot. To what extent is their microbiome reflecting this highly trained state?

2.     Methods, 16S sequencing: which negative controls were included in the analysis and how were reads from the NC handled in the analysis?

3.     Methods, SCFA analysis: the SCFA were only measured in faeces, not in blood. If there was an effect of SCFA on the performance, it would have been very interesting to have the SCFA concentrations in blood in order to see how much of the SCFA are absorbed rather than excreted. Why was this not done?

4.     Methods, statistical analysis: given that microbiota data in principle has a high degree of true 0 values, why was the analysis not fully conducted using non-parametric statistics?

5.     Results, alpha diversity analysis: Given that the Shannon index includes a measure of both richness and evenness and Simpson again is a measure of evenness, I would recommend either adding a measure of richness (like Chao1) or take out the Simpson results. It seems unbalanced at the moment.

6.     Results, table 1: this table should be expanded to show BMI, the dietary intake of the participants in the different parts of the Vuelta  including overall energy intake, and the intake of macronutrients, body composition of the riders before and after the race.

7.      Results, line 226: if the abundance of Actinobacteria increases, which family is the main responsible family for this (the Coriobacteriaceae)? Have you correlated the abundance of Actinobacteria to specific dietary components?

8.     Results, lines 248-9: I am not sure which figure these lines relate to.

9.     Results, figure 3: please also show the true PCA plots. Currently, you are only showing which families contribute to the components, which is nice but quite meaningless without actually viewing the PCA plots. It would be nice to have all 4 timepoints in the same plot, if possible. This may obviate the use of figure 3B. It may also explain how the family of Bifidobacteroidaceae is present both in PC1 and PC2. Figures 3E and 3Fare difficult to interpret, the top lines can be distinguished but the ones underneath cannot and therefore do not contribute much to the reader.

10.  Results, figure 4: similar comments to the comments for figure 3 hold true for this figure. You should show the plot first so that people can see how the samples (individuals) differ from each other and between timepoints. Then the other figures may become more relevant. Figure 4C which shows the concentrations of SCFA across time, please check the units because it seems very high.

11.  Results, figure 4D: it actually seems that (iso)butyrate and isovalerate vary the most between different components. Again, without the plot showing the individuals across time, it is difficult to interpret this figure.

12.  Results, paragraph lines 287-298/figure 5: I am not an expert but I can imagine that if there are mountain stages in specific parts of the Vuelta that this may affect the correlations described here. Maybe this should be taken into account as well. In addition, the teams also have strategies within the team where some riders support others and thereby help to move others up/keep them at the top. This may therefore affect the correlations observed here. I am not sure if this analysis, which is really more about the inherent qualities of the rider as well as other factors is meaningful here.

13.  Results, figure 6/section gut microbiota dynamics predicts performance: I find this analysis very circular. It is very much possible that it is not the bacteria being correlated to the performance but the factors that explain the difference in bacteria e.g. dietary intake also explain the difference in performance. This seems to be the case in table 3/figure 7

14.  Results, table 3/figure 7 and results text: I think you need to be very careful with these analyses since you only have information on 8 participants. Your wording also needs to more clearly reflect that the p-values actually are not significant for the Bifidobacteriaceae. The question is also whether figure C which in a way based on the same data as B is meaningful to add here (or figure B).

15.  Discussion, line 402/3: I think you need to be very careful with the statement that the microbiota are performance modulators. You have not shown that a direct correlation between some of the factors that determine the changes in the bacteria are not directly correlated with performance.

16.  Discussion, lines 461-463: would the fibre intake while maintained in absolute numbers not decrease dramatically when considered in the context of the significant increase in total calories and especially high GI carbohydrates? This could have significant effects and could perhaps mask the relationship. The same is true for the relationship with the Erysipelotrichaceae.

17. I am missing a section with the limitations of the study.

18.  Discussion, lines 500-502: this is an overinterpretation of your results. Please tone it down.

Round 2

Reviewer 2 Report

Comments and Suggestions for Authors

no more comments

Reviewer 3 Report

Comments and Suggestions for Authors

Thank you to the authors for the revision. I could not see the figures and given that there have been a few updates to the figures, I would like to see these.

The authors have answered all my queries.

Round 3

Reviewer 3 Report

Comments and Suggestions for Authors

Thank you, I have no further queries.